# Non-Ignorable Differences in NIRv-Based Estimations of Gross Primary Productivity Considering Land Cover Change and Discrepancies in Multisource Products

Jiaxin Jin [1,2], Weiye Hou [1], Longhao Wang [1,3,*], Songhan Wang [4,5], Ying Wang [6], Qiuan Zhu [1], Xiuqin Fang [1] and Liliang Ren [1]

1 College of Hydrology and Water Resources, Hohai University, Nanjing 210024, China; jiaxinking@hhu.edu.cn (J.J.); houweiye@hhu.edu.cn (W.H.); zhuq@hhu.edu.cn (Q.Z.); kinkinfang@hhu.edu.cn (X.F.); rll@hhu.edu.cn (L.R.)
2 Key Laboratory of Water Big Data Technology of Ministry of Water Resources, Hohai University, Nanjing 210024, China
3 Key Laboratory of Water Cycle and Related Land Surface Processes, Institute of Geographic Sciences and Natural Resources Research, Chinese Academy of Sciences, Beijing 100101, China
4 Key Laboratory of Urban Land Resources Monitoring and Simulation, Ministry of Natural Resources, Jiangsu Collaborative Innovation Center for Modern Crop Production, College of Agriculture, Nanjing Agricultural University, Nanjing 210095, China; wangsonghan@njau.edu.cn
5 Key Laboratory of Crop Physiology and Ecology in Southern China, College of Agriculture, Nanjing Agricultural University, Nanjing 210095, China
6 Tourism and Social Administration College, Nanjing Xiaozhuang University, Nanjing 211171, China; mfacewang@njxzc.edu.cn
* Correspondence: wanglonghao0857@igsnrr.ac.cn; Tel.: +86-10-64856515

**Abstract:** The accurate estimation of gross primary productivity (GPP) plays an important role in accurately projecting the terrestrial carbon cycle and climate change. Satellite-driven near-infrared reflectance (NIRv) can be used to estimate GPP based on their nearly linear relationship. Notably, previous studies have reported that the relationship between NIRv and GPP seems to be biome-specific (or land cover) at the ecosystem scale due to both biotic and abiotic effects. Hence, the NIRv-based estimation of GPP may be influenced by land cover changes (LCC) and the discrepancies in multisource products (DMP). However, these issues have not been well understood until now. Therefore, this study took the Yellow River basin (YRB) as the study area. This area has experienced remarkable land cover changes in recent decades. We used Moderate-Resolution Imaging Spectroradiometer (MODIS) and European Space Agency (ESA) Climate Change Initiative (CCI) land cover products (termed MCD12C1 and ESACCI, respectively) during 2001–2018 to explore the impact of land cover on NIRv-estimated GPP. Paired comparisons between the static and dynamic schemes of land cover using the two products were carried out to investigate the influences of LCC and DMP on GPP estimation by NIRv. Our results showed that the dominant land cover types in the YRB were grassland, followed by cropland and forest. Meanwhile, the main transfer was characterized by the conversion from other land cover types (e.g., barren) to grassland in the northwest of the YRB and from grassland and shrubland to cropland in the southeast of the YRB during the study period. Moreover, the temporal and spatial pattern of GPP was highly consistent with that of NIRv, and the average increase in GPP was 2.14 $gCm^{-2}yr^{-1}$ across the YRB. Nevertheless, it is shown that both LCC and DMP had significant influences on the estimation of GPP by NIRv. That is, the areas with obvious differences in NIRv-based GPP closely correspond to the areas where land cover types dramatically changed. The achievements of this study indicate that considering the land cover change and discrepancies in multisource products would help to improve the accuracy of NIRv-based estimated GPP.

**Keywords:** NIRv; GPP; land cover; MCD12C1; ESA CCI LC maps; Yellow River basin

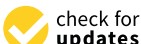



## 1. Introduction

Gross primary productivity (GPP) is the total amount of organic matter produced by terrestrial plants through photosynthetic uptake of atmospheric $CO_2$ [1,2]. It plays an important role in the global carbon cycle and has a significant impact on mitigating climate change [3,4]. Enhancing GPP is an important way to achieve sustainable development [5], and consequently, the accurate estimation of GPP is of vital importance in the current changing world [6]. At present, GPP has been widely quantified by different methods, e.g., estimations from the machine-learning upscaling approach [7,8], light-use-efficiency (LUE) models [9,10], and ecosystem processed-based models [11–13]. However, due to the complex models and many influencing factors, the quantification of GPP still shows considerable uncertainties [14,15].

Recent advances in satellite-based direct proxies for GPP, such as near-infrared reflectance (NIRv), provide new opportunities for the accurate estimation of GPP on a large scale [16,17]. As the product of the normalized different vegetation index (NDVI) and NIR reflectance [16], NIRv is easily acquired. Moreover, NIRv can effectively isolate vegetation signals and reduce the effect of soil noise on the vegetation spectrum [16]. Previous studies have shown that NIRv has a strong correlation with GPP at different time scales [17,18], and NIRv could be used for global GPP estimation with long time series [19–21]. Compared with traditional models of GPP quantification, using NIRv as a proxy to estimate GPP is simpler, and it does not need to input environmental information, such as temperature, precipitation, etc. [21,22].

Notably, as one of the most important processes of global environmental change, land cover (LC) has important impacts on GPP estimation [23]. The effects of land cover on GPP can be discussed from two aspects, namely, the dynamic land cover change (LCC) and the discrepancies in multisource products (DMP) [24,25]. On the one hand, LCC is the most important driving force of land surface processes, and it has an especially profound impact on the global carbon cycle [26]. With changes in time, the land cover types (LCTs) of pixels may change, which would lead to uncertainties in GPP estimation. In theory, GPP quantification by NIRv would be highly dependent on LCTs because of NIRv's simple parameters (linear parameters for different land cover) [22,26]. Taking the LC of a certain year as the predictive parameter, Wang et al. used NIRv to dynamically estimate GPP based on different land cover types, and the results were satisfactory [22]. However, the impact of LC on the relationship between GPP and NIRv has not been systematically explored. In previous studies focusing on the estimation of GPP by NIRv, little attention has been paid to the parameters of land cover and its changes.

On the other hand, the discrepancies in multisource products would cause differences in the expression of land cover heterogeneity [27], which would further influence the GPP estimation based on NIRv. Accurately capturing the GPP changes caused by LCC requires high-quality land cover data [26]. Previous studies usually used the Moderate-resolution Imaging Spectroradiometer (MODIS) land cover product classified by the International Geosphere Biosphere Programme (IGBP) (MCD12C1) with long time series and various land cover types. This product, however, has the problem of relatively rough resolution. The study of Cracknell et al. pointed out that the accuracy of the quantization of GPP could be improved by using high-resolution land cover maps [28]. The European Space Agency (ESA) Climate Change Initiative (CCI) Maps (ESACCI) show a high accuracy and resolution [29]. In theory, due to the great differences between MCD12C1 and EASCCI, the GPP estimations based on these two products would also be different [27,30]. However, for the emerging method of estimating GPP by NIRv, few studies systematically pay attention to the impact of land cover products.

The goal of this study is to evaluate the influences of LCC and DMP on NIRv-based GPP estimation and to explore the corresponding uncertainties. Therefore, we took the Yellow River basin in China, where land cover has changed drastically in recent decades, as the study area, and chose two different land cover products (i.e., MODIS and ESACCI) to explore the distribution of and change in land cover from 2001 to 2018. Firstly, we set up

the static and dynamic schemes of LCC and the comparative experiments of different land cover products. Then, we derived annual average GPP maps and temporal and spatial changes across the Yellow River basin and its sub-basins. Finally, we clarified the influences of LCC and DMP on GPP estimation from NIRv. We aimed to point out that it is necessary to consider land cover changes and products when using NIRv to estimate GPP.

## 2. Materials and Methods

### 2.1. Study Area

The Yellow River basin (YRB) ($32°-42°$N and $96°-119°$E, Figure 1), which is one of the most important watersheds across north China, was selected as the study area for its outstanding vegetation cover change [31,32]. Its total watershed size is about 79,500 km$^2$, with a quite developed river system. The YRB encompasses a complex terrain that spans a large elevation range, e.g., the Qinghai-Tibet Plateau, the Loess Plateau, and the North China Plain. It contains eight sub-watersheds, i.e., the area above Longyang Gorge (ALG), Longyang Gorge to Lanzhou (LGL), Lanzhou to Hekou Town (LHT), Hekou Town to Longmen (HTL), Longmen to Sanmen Gorge (LSG), Sanmen Gorge to Huayuankou (SGH), the area below Huayuankou (BH), and an inflow zone (IZ). The YRB is a typical transitional zone between a continental climate in the western part and a monsoon climate in the eastern part. It is covered by numerous kinds of vegetation with the diverse climate, dominated by grass followed by crops and mixed forests.

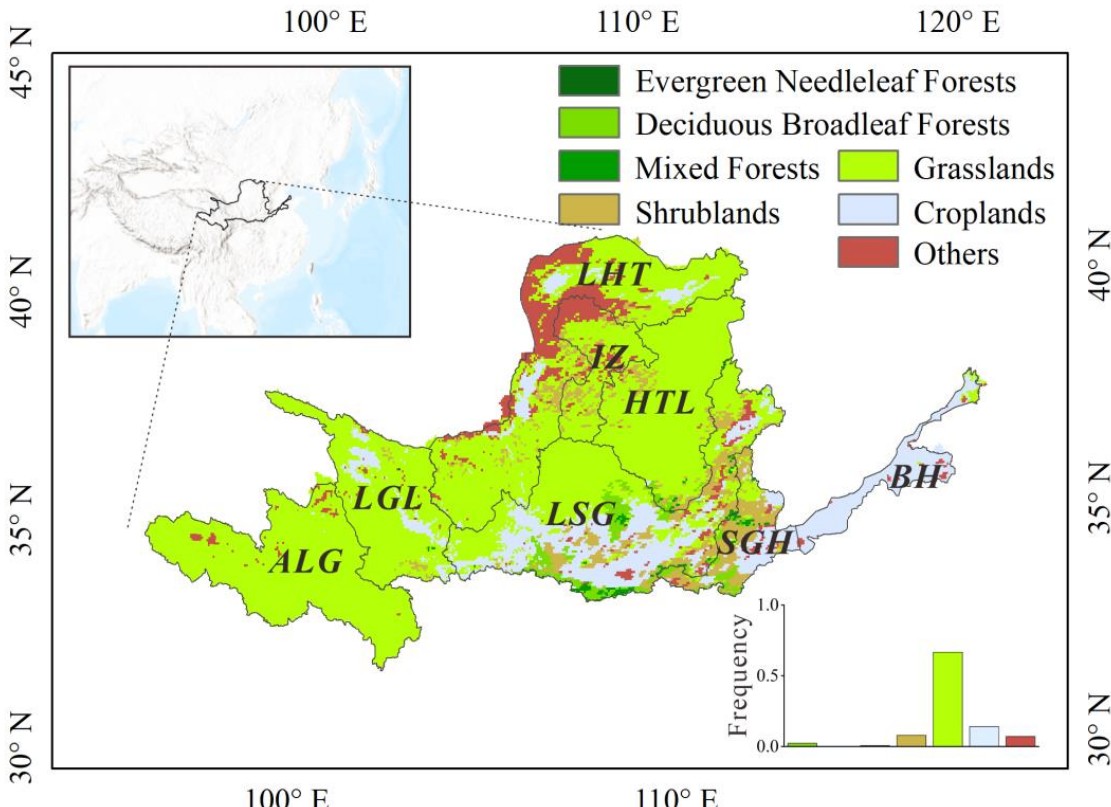

**Figure 1.** Spatial pattern of MCD12C1 land cover over the Yellow River basin (YRB) in 2001. Only natural plants (the colored areas) were included in this study. The sub-basins of the YRB include the area above Longyang Gorge (ALG), Longyang Gorge to Lanzhou (LGL), Lanzhou to Hekou Town (LHT), Hekou Town to Longmen (HTL), Longmen to Sanmen Gorge (LSG), Sanmen Gorge to Huayuankou (SGH), the area below Huayuankou (BH), and an inflow zone (IZ). Frequency histogram displaying the areal proportions (%) of corresponding land cover types is inset.

The YRB has experienced dramatic changes in the landscape because of intensive climate change and increasing human activities during the past decades [33]. Several con-

servation programs, the Grain for Green (GFG) project and the Natural Forest Conservation (NFC) program, have been carried out to mitigate serious environmental and ecological issues [34–37]. Land surface greening has been widely observed across the YRB since 2000 [38,39]. It is reported that the vegetation coverage on the Loess Plateau, especially, has doubled from 31.6% in 1999 to 59.6% in 2013 [40]. Hence, the YRB is regarded as a natural laboratory for studying land cover change and its corresponding effects [31,32].

### 2.2. Land Cover Data and Vegetated Grid Cells of Interest

The multisource land cover products used in this study were MCD12C1 and ESACCI, representing qualitative and semiquantitative land cover products according to Wang and Jin [30]. The MCD12C1 land cover products describe the land cover properties derived from observations of Terra and Aqua MODIS data with a spatial resolution of 0.05° and the span of year, which can be downloaded at https://lpdaac.usgs.gov/products/mcd12c1v061/ (accessed on 29 August 2023). The product has been widely used to quantify global land cover with long time series. We used the IGBP global vegetation classification scheme with 17 land cover classes [41]. The ESACCI is a representative global semi-quantitative land cover product with a spatial resolution of 300 m, which can be obtained from http://maps.elie.ucl.ac.be/CCI/viewer/download/ (accessed on 22 December 2019). The product provides a global map that divides the land surface into 37 categories, which are defined by the Land Cover Classification System (LCCS) of the Food and Agriculture Organization of the United Nations (FAO). The data show a high accuracy for the types of city, paddy field, and evergreen broad-leaved forest, and they have been widely used to analyze land cover transitions [30,42]. Because this study aimed to quantify the proportion of plant functional types (PFTs), we summarized the ESACCI data with the 37 categories into grid cells at a 0.05° spatial resolution (consistent with MODIS) and converted them into PFTs [43] for the subsequent work.

Given the natural vegetation in the study area, the proportion of each PFT was calculated through the PFT cross-walking table, as shown in Table 1. This study used a PFT-based ESACCI land cover product to explore LCC and estimate GPP. PFT can better determine the ecological process as well as detect and predict the vegetation response to a series of environmental changes [44,45]. The ESACCI PFT-based land cover products were converted from qualitative pixel types to land cover percentages. When exploring the relationship between GPP and NIRv, the estimated GPP was weighted and summed according to the corresponding PFTs.

**Table 1.** Land cover products used in this work.

| MODIS | ESA CCI (PFTs) |
| :---: | :---: |
| Deciduous broadleaf forest | Deciduous broadleaf tree |
| / | Evergreen broadleaf tree |
| Evergreen needleleaf forest | Evergreen needleleaf tree |
| Mixed forest | / |
| Closed shrub | Evergreen broadleaf shrub |
| | Deciduous broadleaf shrub |
| Sparse shrub | Evergreen needleleaf shrub |
| | Deciduous needleleaf shrub |
| Grass | Grass |

In the study, we excluded bare ground through NDVI < 0.1, because it will not make any contribution to GPP [46]. There were croplands in the study area, including C3 and C4 croplands. Since this study only focused on the differences caused by land cover rather than specific species, only C3 cropland was taken as a proxy of cropland to calculate GPP.

### 2.3. NIRv and GPP Calculations

A long-term dataset of the Advanced Very-High Resolution Radiometer (AVHRR) reflectance at the red band (R) and near-infrared band (NIR) was applied to calculate the monthly NIRv over the YRB from 2001 to 2018. The dataset was derived from the Land Long-Term Data Record (LTDR) product (v4) at a spatial resolution of 0.05°. The LTDR product performs well in the sensor post-launch calibration, as well as for bias corrections for systematic orbit shifts, cloud screening, sensor degradations, and atmospheric corrections [47].

The cloudy and cloud shadow pixels were excluded in this work, and the median value of the reflectance of all the available pixels was obtained by month for each of the grid cells of interest. Then, monthly NIRv was calculated by the following equation:

$$NIRv = NIR \times (NDVI - 0.08) \tag{1}$$

$$NDVI = (NIR - R)/(NIR + R) \tag{2}$$

A constant of 0.08 was subtracted from all NDVI values to minimize the effect of bare soil [16,22]. Moreover, the NIRv values that were less than or equal to zero were excluded in the following analysis [16].

The NIRv–GPP relationships proposed by Wang et al. were used to estimate GPP for the different land cover types in this study [22]. For the percentage of covered PFTs, we calculated the weighted average of GPP. The relationships were established using the GPP estimations and satellite NIRv at 104 eddy-covariance (EC) flux sites. The linear regression between GPP and NIRv was calibrated and validated for each land cover type (Table 2). The models performed well across all of the types, with an average $R^2$, bias, and RMSE of 0.71, 0.02, and 1.95 gC m$^{-2}$ d$^{-1}$, respectively. More details of the methods and evaluations are described in Wang et al. [22]. Notably, large differences in the parameters of the linear models were observed across the vegetation types, which indicated the relationships between GPP and NIRv were likely to be biome-specific [21]. That is why the schemes of land cover change and the discrepancy in multisource products proposed in this study should be considered in NIRv-based GPP estimation.

**Table 2.** Coefficients and overall performances of the linear models between NIRv and GPP proposed by Wang et al. [22]. $R^2$, root mean square error (RMSE), and bias values are presented for each vegetation type.

| Land Cover Type | Linear Regression | $R^2$ | RMSE | Bias |
|---|---|---|---|---|
| DBF | GPP = 64.07·NIRv − 2.20 | 0.65 | 2.14 | 0.14 |
| EBF | GPP = 44.50·NIRv + 2.60 | 0.45 | 2.02 | 0.56 |
| ENF | GPP = 64.51·NIRv − 1.41 | 0.65 | 1.96 | 0.10 |
| MF | GPP = 59.49·NIRv − 2.93 | 0.70 | 1.86 | 0.04 |
| SHR | GPP = 36.18·NIRv − 0.87 | 0.70 | 0.72 | −0.03 |
| GRA | GPP = 68.13·NIRv − 1.62 | 0.74 | 1.94 | 0.05 |
| CRO-C3 | GPP = 55.38·NIRv − 1.97 | 0.65 | 2.14 | 0.14 |

### 2.4. Statistical Analysis

The land cover changes were investigated over the YRB from 2001 to 2018 using transfer matrix analysis for the MODIS and CCI land cover maps. A fraction-based transfer matrix approach, in particular, was applied to the CCI data with the PFTs [48,49]. The approach assumed that there was no transition between the same PFT, and the transferred amount among different PFTs depended on the missing amount of each PFT for two consecutive years. More details of the method are described in Ji et al. [48].

We cross-compared the GPP estimations over the YRB, respectively, using static land cover schemes across the entire study period (termed staMOD and staCCI for the MODIS and CCI data, respectively) and dynamic land cover schemes (termed dynMOD and

dynCCI for the MODIS and CCI data, respectively). For the static schemes, the coefficients of the NIRv–GPP model were treated as constant values across the entire study period for each pixel, which were assigned by the land cover map in 2001. For the dynamic schemes, the coefficients of the model were treated as annual varying variables parameterized from the annual land cover type for each pixel. The cross-model comparison was conducted at the pixel and sub-basin levels. All of the models using the static/dynamic LC schemes were driven by the same NIRv data. Hence, the GPP difference between the two static schemes could be attributed to the difference between the multisource products, and that between the static and dynamic schemes resulted from the land cover change.

The Theil–Sen estimator was adopted to detect the inter-annual changes in land cover fraction, NIRv, and modeled GPP by pixel across the vegetated area of interest. The Theil–Sen estimator is a nonparametric method with a better robustness to outliers [50]. Its significance was evaluated by the Mann–Kendall trend test. The Kruskal–Wallis test was used to study the reliability of a result by comparing the result with other products. The test is a nonparametric method to investigate whether two or more groups of continuous or discrete variables come from the same probability distribution [51]. The Bonferroni–Dunn test was used to calculate the Dunn's *p*-value after performing the Kruskal–Wallis test analysis [52,53].

## 3. Results

### 3.1. Land Cover Change

The land cover maps of the study area in 2001 (static) and 2018 (dynamic) were compared using the MCD12C1 data, as shown in Figure 2a,b below. The main land cover type in the YRB was grassland, followed by cropland. Figure 2c,d shows the grid cells with the changes in land cover types between the two years. In 2001, grassland accounted for 67.5% of the whole basin, cropland accounted for 14.3%, shrubland accounted for 8.0%, and forests (DBF, MF et al.) accounted for 2.3%. In 2018, grassland accounted for 67.3% of the basin, cropland accounted for 20.7%, forests (DBF, MF et al.) accounted for 7.4%, and shrubland accounted for 2.5%. Meanwhile, we found that the changes in land cover types were mainly concentrated in the northwest and the southeast of the YRB (Figure 2c,d). Among all of the changed pixels, grassland accounted for 34.7% in 2001 and 33.4% in 2018, which implies that more grassland pixels have been converted into other types. Cropland accounted for 7.1% of the changed pixels in 2001 and for 41.7% of them in 2018, which implies that more other types of pixels were converted into cropland. We found that the emerging grassland pixels were mainly concentrated in the northwest of the study area, and most of them were converted from other types of land (including barren, cities, etc.) and shrubland. The loss of grasslands was mainly distributed in the southeast of the YRB, and most of them have been converted into cropland. During 2001–2018, the number of shrubland pixels also decreased significantly, and most of them were converted into cropland and forests. The main transfer was characterized by the conversion from other types to grassland in the northwest of the YRB and from grassland and shrubland to cropland in the southeast of the YRB.

Figure 3 shows the land cover change rates in the eight sub-basins of the YRB. The circular rings represent the proportion of the land cover transition for DBF, SHR, GRA, and CRO. Overall, DBF and SHR decreased slightly, and GRA and CRO decreased significantly in the YRB. DBF decreased at a rate of $-1.05 \times 10^{-3}$%/yr ($p < 0.05$), and SHR decreased at a rate of $-4.68 \times 10^{-3}$%/yr ($p < 0.05$). CRO decreased by $-1.10 \times 10^{-3}$%/yr, and GRA showed the highest decline rate of $-2.12 \times 10^{-2}$%/yr ($p < 0.05$) during 2001–2018. The land cover transition varied across different sub-basins. LGL and ALG are situated in the upper reaches of the YRB, located in the Qinghai-Tibet Plateau. The primary land cover type of LGL and ALG is GRA, with a low proportion of land cover pixels undergoing change. The land covers of the LHT, IZ, HTL, and LSG, which are located in the middle reaches of the YRB (i.e., the Loess Plateau), changed significantly. Similarly, the changes in

pixel proportion in the BH and SGH, which are located in the lower reaches of the YRB (i.e., the North China Plain), were quite dramatic.

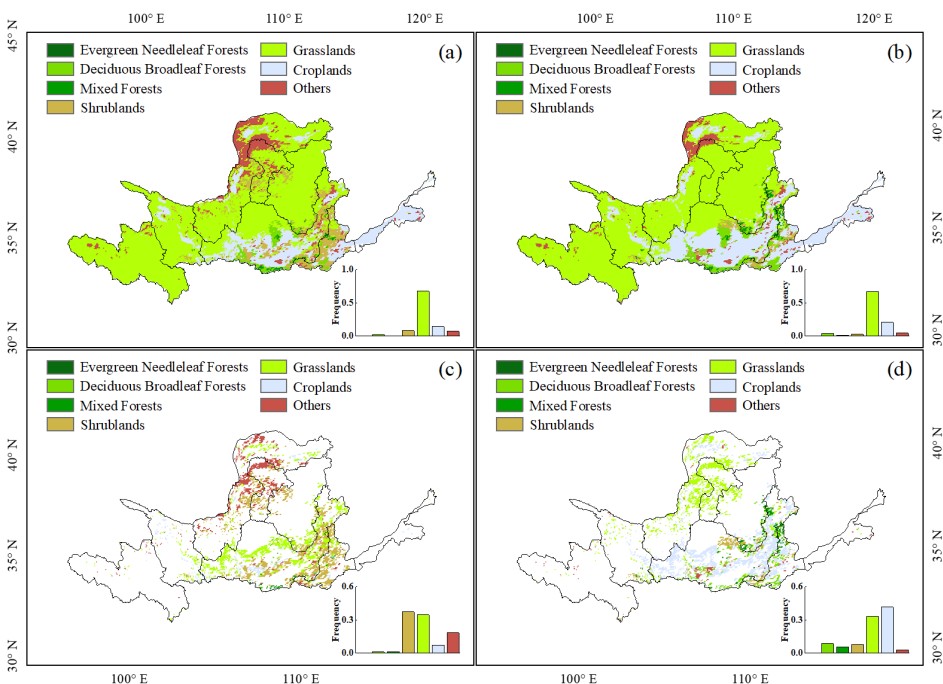

**Figure 2.** The spatial pattern of the land cover types and changes over the Yellow River basin based on the MCD12C1 maps. The panels (**a**,**b**) show the geographical distribution of the different land cover types in 2001 and 2018, respectively. The panels (**c**,**d**) show the land cover types of the changing pixels in 2001 and 2018, respectively. Frequency histograms displaying the areal proportions (%) of corresponding differences are inset.

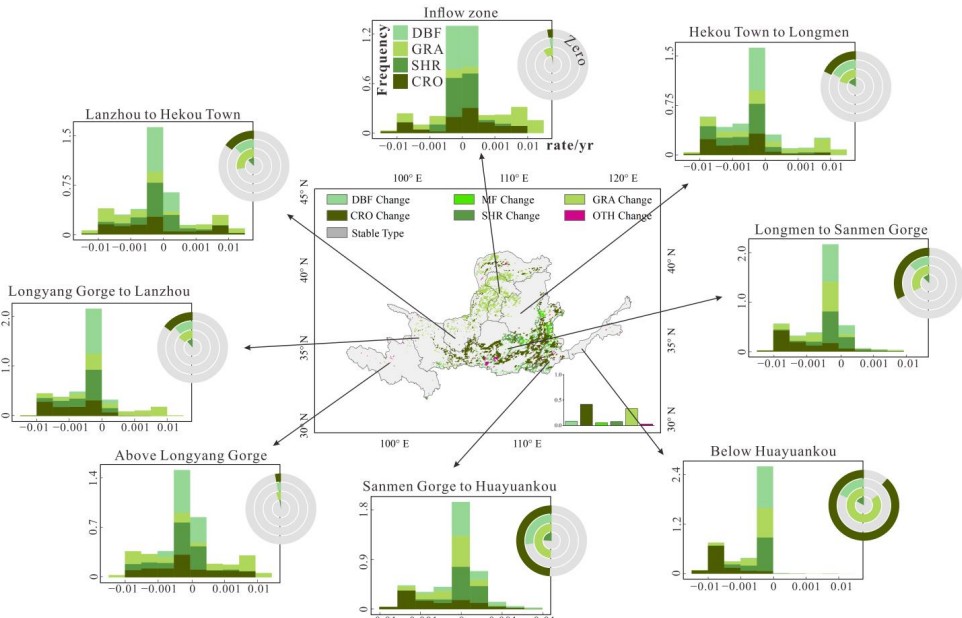

**Figure 3.** Land cover change rate of sub-basins in YRB according to ESACCI. The colored sections on the spatial map represent the land cover transition types that occurred in 2018 (corresponding to the types in Figure 1 from 2001), and there are no land cover changes in the gray areas. The histograms indicate the frequency of variation tendency for DBF, SHR, GRA, and CRO. The circular rings indicate the proportion of the land cover transition for DBF, SHR, GRA, and CRO.

### 3.2. GPP Estimation under the Different Schemes of Land Cover

The NIRv-based GPP can be estimated after clarifying the land cover types and land cover changes in the YRB. Figure 4a shows the GPP estimated by the MODIS product in the year of 2001 (StaMOD). After that, we estimated the difference in GPP caused by dynamic changes in land cover, i.e., the difference between the MODIS-based GPP in 2001 and 2018 (Figure 4b). Then, we estimated the difference in GPP caused by heterogeneous land cover data, i.e., the difference between the MODIS-based GPP and the CCI-based GPP (Figure 4c,d). The average value of GPP in the YRB was 334.16 $gCm^{-2}yr^{-1}$. In addition, the basins with higher GPP were ALG (673.23 $gCm^{-2}yr^{-1}$) and LGL (606.35 $gCm^{-2}yr^{-1}$), whereas those with lower GPP were LHT (45.25 $gCm^{-2}yr^{-1}$) and IZ (5.90 $gCm^{-2}yr^{-1}$). Overall, both LCC and DMP would have significant influences on the estimation of GPP by NIRv. The dynamic changes in land cover over the 18 years resulted in an average increase by 1.40 $gCm^{-2}yr^{-1}$ of GPP. The sub-basins with severe land cover changes were LHT, LSG, and SGH, located in the Loess Plateau in the middle reaches of the YRB (Figure 4b). The results showed that the areas with obvious differences in GPP corresponded to the areas where land cover types changed during 2001–2018 (Figure 2c,d and Figure 4b). The impact of discrepancies in multisource products on the estimation of GPP was significant (Figure 4c), with an average difference of 46.45 $gCm^{-2}yr^{-1}$, which was much greater than that in the GPP estimation caused by the land cover dynamic changes.

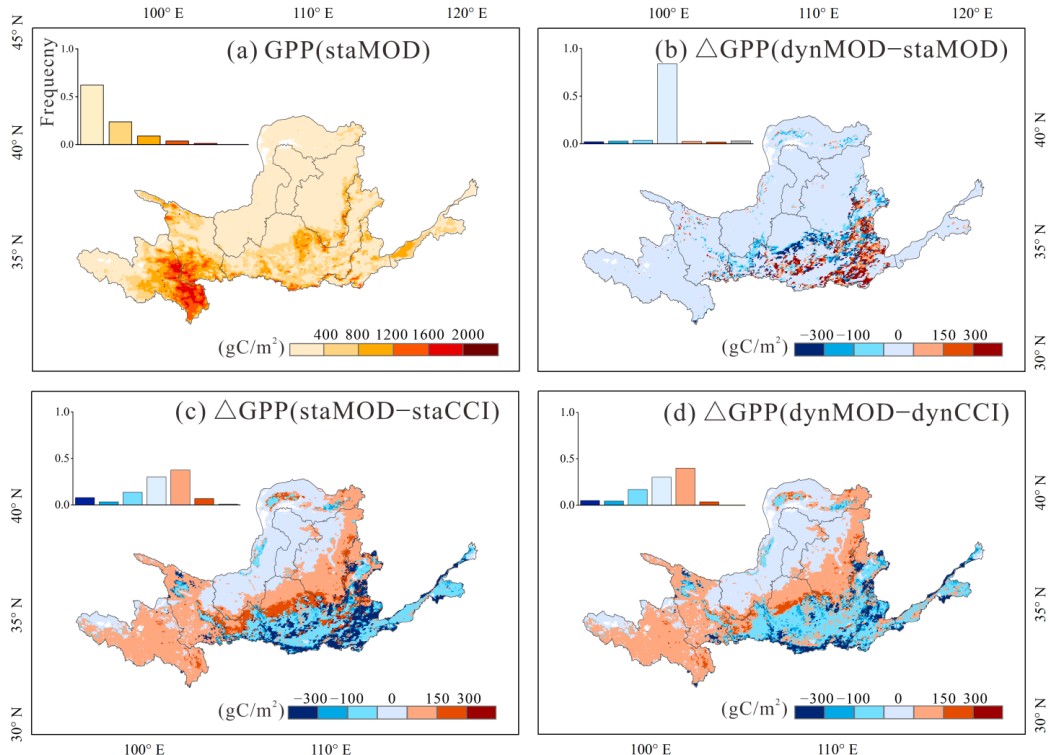

**Figure 4.** Differences in the multi-year average values of NIRv-based GPP using the different land cover (LC) schemes over the YRB from 2001 to 2018. The panel (**a**) shows the geographical distribution of the GPP using MODIS LC data in 2001 (staMOD). The panel (**b**) shows the geographical distribution of the difference between the GPP using annual MODIS LC data from 2001 to 2018 (dynMOD) and that using the MODIS LC data in 2001 (staMOD). The panel (**c**) shows the geographical distribution of the difference between the GPP using MODIS LC data in 2001 (staMOD) and that using CCI LC data in 2001 (staCCI). The panel (**d**) shows the geographical distribution of the difference between the GPP using annual MODIS LC data from 2001 to 2018 (dynMOD) and that using annual CCI LC data from 2001 to 2018 (dynCCI). Frequency histograms displaying the areal proportions (%) of corresponding differences are inset.

The statistics for the multi-year average values of GPP using the different LC schemes and their differences by the sub-basins are shown in Figure 5. Each panel in Figure 5 corresponds to Figure 4. Figure 5a shows the sub-basin statistics of the average staMOD GPP. Figure 5b–d represents the distribution of the differences in GPP. Most of the dynamic GPP changes in sub-basins were positive, i.e., the dynamic changes in land cover over the 18-year period led to an increase in GPP. Figure 4c,d shows the difference in GPP caused by land cover heterogeneity, i.e., the difference between the MODIS-based GPP and the CCI-based GPP. The impact of differences in land cover products on the estimation of GPP was significant, with a high uncertainty. In the middle reaches of the YRB, including HTL and LSG, the average differences between the GPP estimations using the two land cover products were 48.40 gCm$^{-2}$yr$^{-1}$ and 66.24 gCm$^{-2}$yr$^{-1}$, respectively. The aforementioned regions are located in the Loess Plateau, with a high heterogeneity of land cover types that were significantly influenced by the mixed pixels. In the ALG and LGL of the Tibetan Plateau, the differences in GPP derived from the different land cover products were relatively low, with average values of 25.48 gCm$^{-2}$yr$^{-1}$ and 2.31 gCm$^{-2}$yr$^{-1}$, respectively. This is due to the predominant land cover type of GRA in these areas, with a consistent expression between the two products.

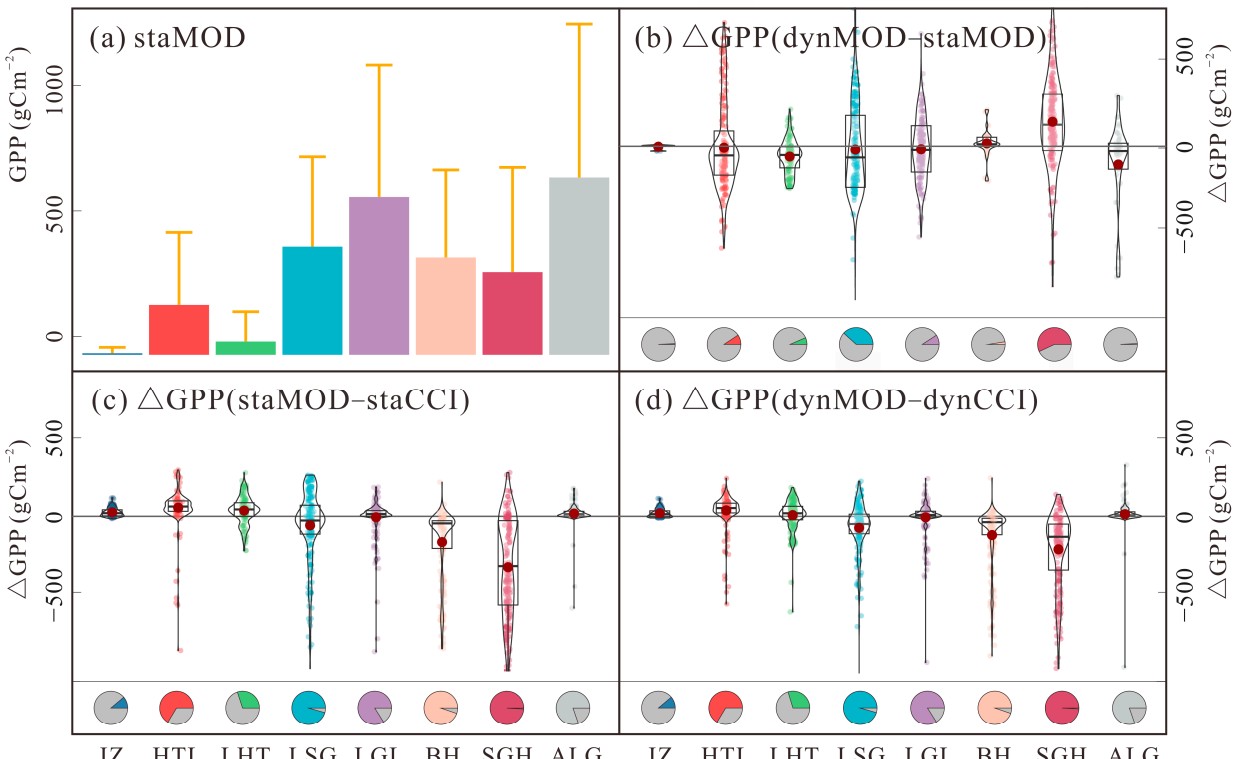

**Figure 5.** Statistics for the multi-year average values of GPP using the different LC schemes and their differences by the sub-basins of the YRB (shown in Figure 1) from 2001 to 2018. The panel (**a**) shows the geographical distribution of the GPP using MODIS LC data in 2001 (staMOD). The panel (**b**) shows the geographical distribution of the difference between the GPP using annual MODIS LC data from 2001 to 2018 (dynMOD) and that using the MODIS LC data in 2001 (staMOD). The panel (**c**) shows the geographical distribution of the difference between the GPP using MODIS LC data in 2001 (staMOD) and that using CCI LC data in 2001 (staCCI). The panel (**d**) shows the geographical distribution of the difference between the GPP using annual MODIS LC data from 2001 to 2018 (dynMOD) and that using annual CCI LC data from 2001 to 2018 (dynCCI). The dark red dots denote median values, the boxes cover the interquartile range, and the thin lines reach the 5th and 95th percentiles. Pie charts displaying the areal proportions (%) of the grid cells with △GPP ≠ 0 in the sub-basins are inset.

### 3.3. Difference in Trends of GPP Derived from NIRv

The estimation of GPP is influenced by land cover changes and related products, and thereby, so is the estimated trend of GPP changes. GPP is the linear product of NIRv based on land cover types. Therefore, the estimation of GPP not only directly depends on land cover types, but it also depends on NIRv. Figure 6a shows the changes in NIRv. The average value of NIRv trends in the YRB was $1.3 \times 10^{-4}$/yr. The overall trend in the YRB was greening. Figure 6b shows the staMOD GPP trends, with an average increase of 2.13 $gCm^{-2}yr^{-1}$. The trend of GPP was highly consistent with the trend of NIRv. For example, the increased regions of GPP were consistent with the increase in NIRv. Figure 6c shows the dynMOD GPP trends, with an average increase of 2.14 $gCm^{-2}yr^{-1}$. Figure 6d shows the dynCCI GPP trends, with an average increase of 1.70 $gCm^{-2}yr^{-1}$. The value was slightly lower than that of MODIS-based GPP trends.

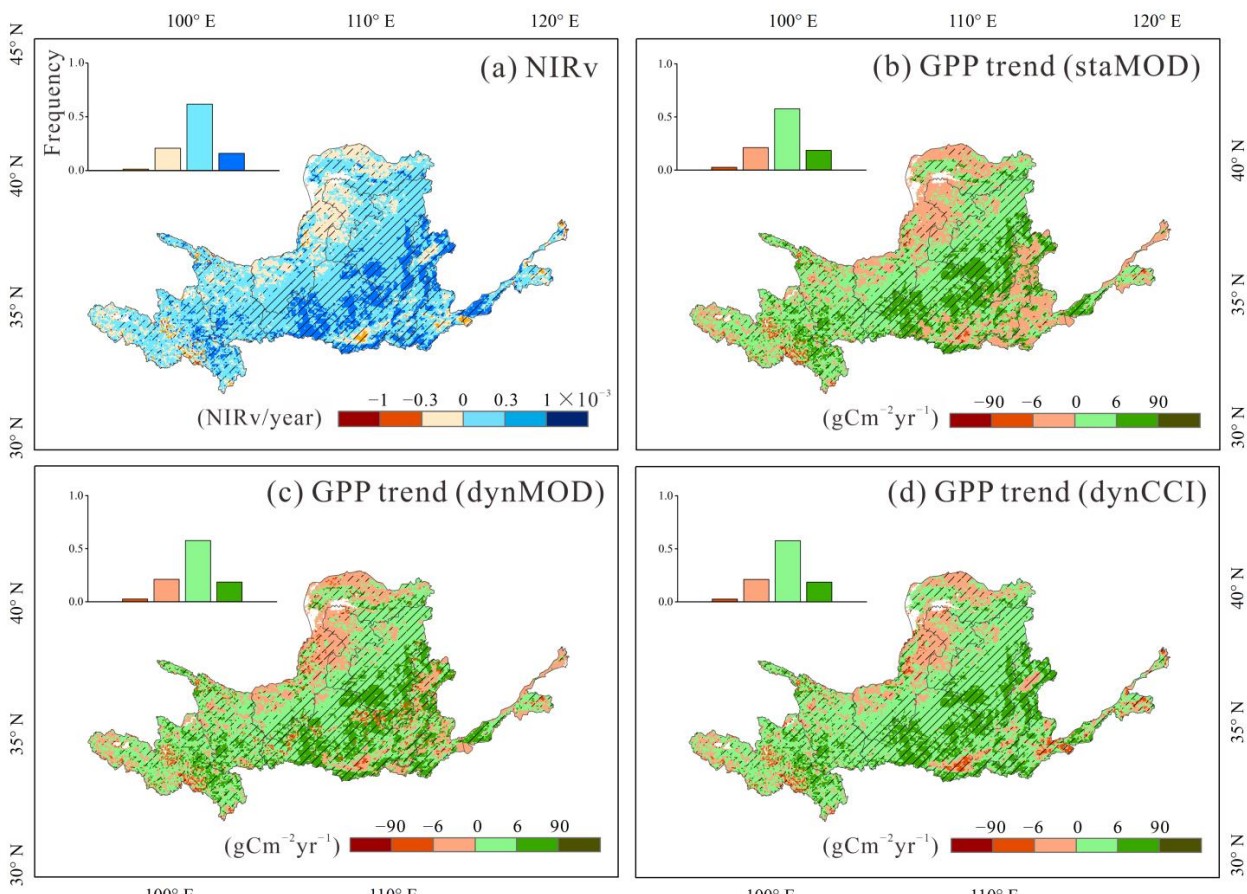

**Figure 6.** Changes in (**a**) NIRv and (**b**–**d**) GPP using the different land cover (LC) schemes, using the Theil–Sen estimator, over the Yellow River basin from 2001 to 2018. The LC schemes include (**b**) the static MODIS LC data in 2001, (**c**) dynamic MODIS LC data, and (**d**) dynamic CCI LC data. The significance of the Theil–Sen slope was estimated by the Mann–Kendall trend test. Shaded regions indicate where changes were significant at *p* < 0.05. Frequency histograms displaying the areal proportions (%) of corresponding changes are inset.

The statistics for the multi-year average trends of GPP using the different LC schemes and their differences by the sub-basins are shown in Figure 7. Figure 7a shows the NIRv trends, with the highest increase of $2.3 \times 10^{-4}$/yr in the LSG. This basin was located in the Loess Plateau, and the effects of GFG were significant. In contrast, ALG had the lowest NIRv trends, with an average of $0.2 \times 10^{-4}$/yr. This basin was located in the Qinghai-Tibet Plateau, and the primary land cover was GRA with barely any changes. The trends of GPP were highly consistent with that of NIRv, with a similar distribution represented in the box

plots. Figure 7b shows that there is little difference in the trends of dynamic and static GPP, and the impact of multisource products' discrepancies on GPP trends is relatively obvious. The differences caused by the discrepancies in multisource products seemed to be larger than those caused by land cover dynamic changes.

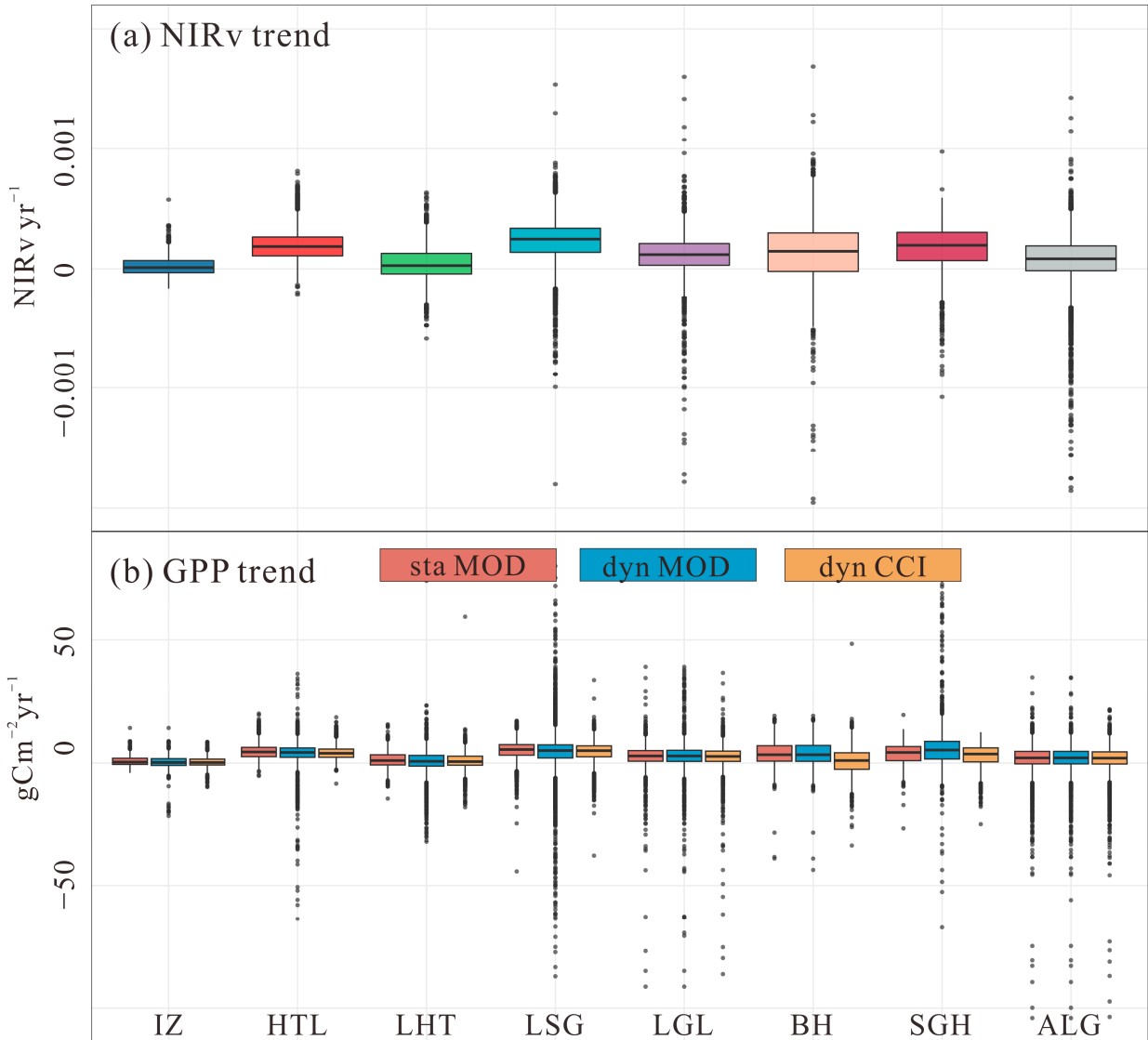

**Figure 7.** Statistics for the changes in (**a**) NIRv and (**b**) GPP in the different land cover (LC) schemes, using the Theil–Sen estimator, by the sub-basins of the Yellow River basin (shown in Figure 1) from 2001 to 2018. The LC schemes include the static MODIS LC data in 2001 (in red), dynamic MODIS LC data (in green), and dynamic CCI LC data (in blow) in the panel (**b**). The red dots denote median values, the boxes cover the interquartile range, the thin lines reach the 5th and 95th percentiles, and the black points denote the outliers.

## 4. Discussion

### 4.1. Relationship between NIRv and GPP

As the product of NDVI and NIR, NIRv has been proven to have strong and linear correlations with GPP in various ecosystems and different time scales [17,18,21]. Wang et al. used NIRv to dynamically estimate GPP based on different land cover types and spatial distributions, and the seasonal patterns of their results were consistent with some mainstream GPP products [22]. However, due to the simple linear relationship between them, in theory, the GPP estimation based on NIRv should be highly dependent on LC (Figure 8c).

Moreover, the multisource products' discrepancies would have an important impact on the estimation of GPP by NIRv (Figure 8b). These issues provided the basis and hypothesis for this study.

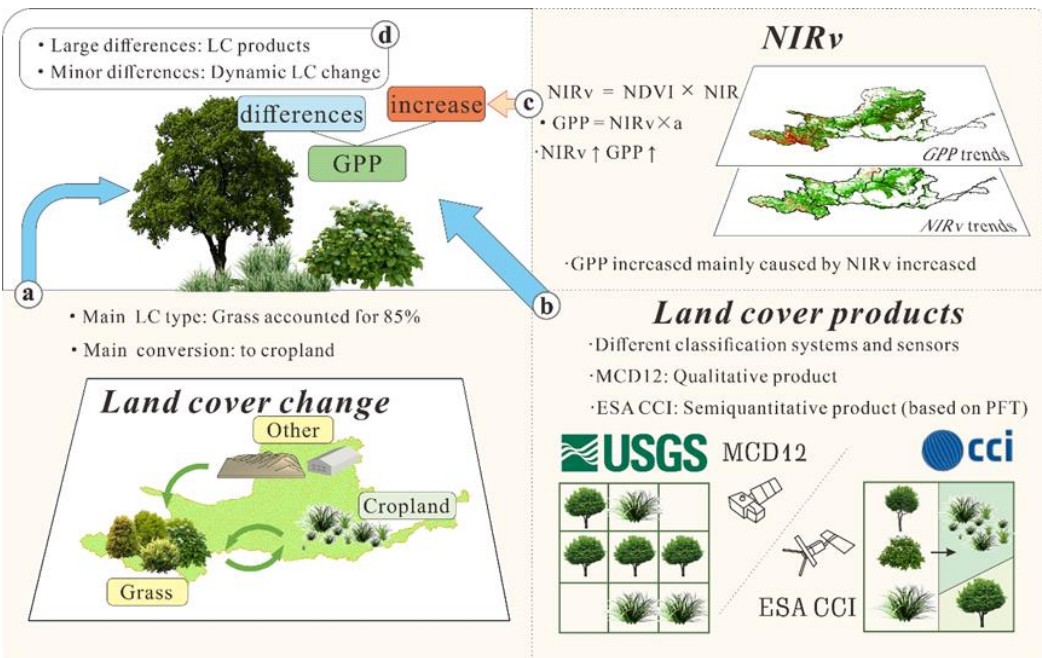

**Figure 8.** The conceptual illustration of the impacts of land cover change (**a**) and the multisource products' differences (**b**) on the estimation of GPP by NIRv (**c**) and their relative contribution (**d**).

The temporal and spatial variations in NIRv in the study area were highly consistent with that of GPP. The changes in GPP were apparently driven by the changes in NIRv. Meanwhile, the coefficients of the linear relationship between GPP and NIRv depend on land cover types [22]. In order to reflect the influence of LCC on GPP estimation, we designed dynamic and static schemes of LC. The results showed that even if NIRv was the same, the estimated GPP based on different land cover products was still quite different. Our results showed that the influence of LCC and DMP on the relationship between GPP and NIRv was obvious and important (Figure 2c,d and Figure 3b–d), and the trend of NIRv without considering land cover change was not a substantial GPP trend (Figure 6), which confirmed our hypothesis. Zhang et al. investigated the effects of LCC on the annual GPP trend over the conterminous United States and showed the influences were significant, which also supports our results [26].

### 4.2. Reliability of the Estimated GPP

Eight extra GPP products were used in our study to validate the reliability of our results. As illustrated in Figure 9, the four schemes of NIRv-based GPP were nearly identical to seven of the eight products from the North American Carbon Program (NACP) Multi-scale synthesis and Terrestrial Model Intercomparison Project (MsTMIP) presented in the boxplots according to the significance calculated by Kruskal–Wallis and Bonferroni–Dunn post-hoc tests, indicating their robustness. Specifically, the average value of bias was 0.075. The bias of dynamic GPP decreased by 0.002 compared to static GPP. We collated the eddy-covariance (EC) observed monthly GPP as the field observations to show the performance among the schemes. The geographical location of the three EC flux towers is shown in Figure S1, and the validation scatter plots are shown in Figure S2. The results of the reliability suggest that (i) the dynamic schemes had better performances than the static schemes, and (ii) our schemes were robust compared to a wide range of GPP products.

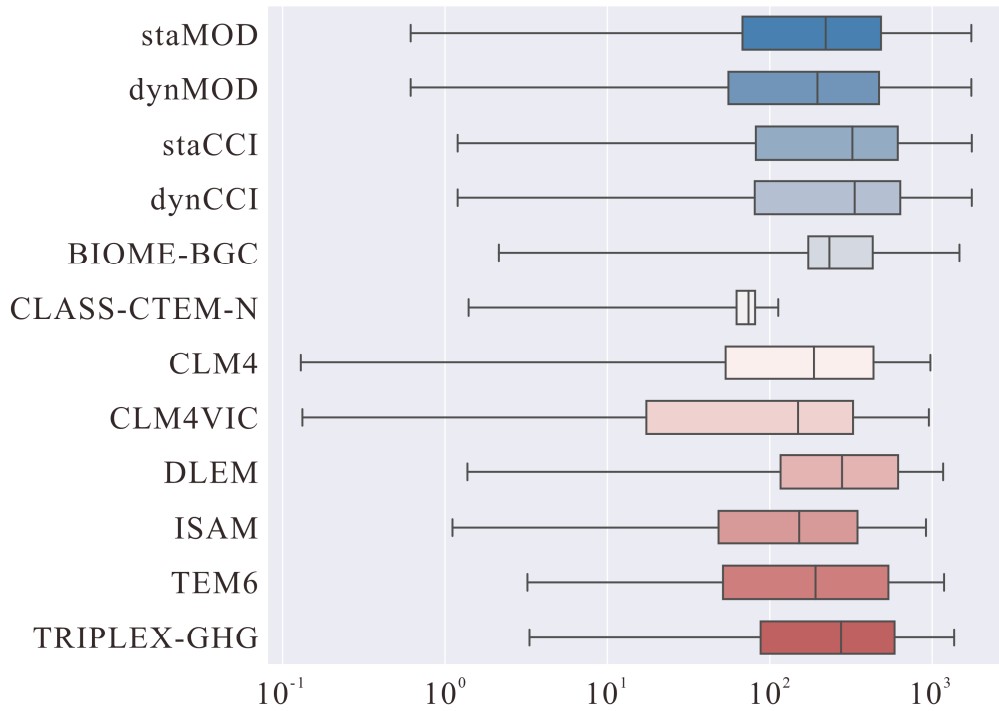

**Figure 9.** Boxplots of GPP distribution obtained from the different models and products. First four blue boxplots are obtained from the NIRv-based GPP with the four schemes (staMOD, dynMOD, staCCI, and dynCCI), respectively. The red boxplots represent the eight different GPP products. For each sub-boxplot, the box plots show the lower whisker (=Q1 − 1.5IQR), lower quartile (Q1), median, upper quartile (Q3), and upper whisker (=Q3 + 1.5IQR), where the IQR = Q3 − Q1, from the bottom to top.

### 4.3. Uncertainties and Further Studies

This study explores the influence of land cover on GPP estimation by NIRv. Using NIRv as a proxy of GPP is a new method, and the influence of land cover change on NIRv-GPP has not been systematically studied. Cracknell et al. thought that the classification system was the reason for the differences between theoretical and measured GPP estimation at coarse resolution [28]. Robinson et al. produced two sets of GPP with different remote-sensing products derived from different sensors, and the results showed significant differences [54]. Our results showed that the dynamic change in LC had a significant impact on the GPP estimation of the Yellow River basin (Figure 5). However, the influence of LC on NIRv-GPP in areas with different intensities of LCC (such as Qinghai-Tibet Plateau) is still not clear. In addition, due to the lack of spatial observation data, we could not determine which land cover product was most in line with the actual land type distribution in the study area. Our work only cross-contrasted the two land cover products (that is, MCD12C1 V6 and ESACCI), which are widespread products and could be well matched with remote-sensing GPP estimation. However, both of the two products show unreasonable phenomena, such as misclassification or mixed pixels. The ESACCI also introduced uncertainties in the scaling transformation. The defects of land cover products might lead to uncertainties in GPP estimation. More products could be considered in the follow-up study to provide references to accurately estimate GPP by NIRv. Notably, the relationship between NIRv and GPP might be influenced by a series of other factors, such as community succession, anthropogenic management, fertilization, etc., which also need to be further explored. At present, our work just gives a clear signal that the LCC and the differences between different products need to be considered in the GPP estimation. In the future, we will conduct more detailed studies on these aspects to give practical suggestions for GPP estimation.

## 5. Conclusions

This study explores the influences of land cover change and multisource products' differences on GPP estimation by NIRv. It is reported that the land cover of the YRB has undergone drastic dynamic changes in recent decades due to the ecological projects. Therefore, this study took the YRB as the study area and used MCD12C1 and ESACCI land cover data during 2001–2018 to explore the impacts of land cover on NIRv-based estimated GPP. Our results showed: (1) The dominant land cover types in the YRB were grassland, followed by cropland and forest. The main land cover transfer was characterized by the conversion from other types of land to grassland in the northwest of the YRB and from grassland and shrubland to cropland in the southeast of the YRB during the study period. (2) The temporal and spatial pattern of GPP was highly consistent with that of NIRv, and the average increase in GPP was 2.14 $gCm^{-2}yr^{-1}$ across the study area. (3) Both LCC and DMP had significant influences on GPP estimated by NIRv. Specifically, the areas with obvious differences in NIRv-based GPP closely corresponded to those where land cover types dramatically changed. Our study thus demonstrates that the influences of land cover on the relationship between NIRv and GPP are non-negligible, and considering the land cover change and differences of multisource products would be beneficial to improve the accuracy of carbon flux estimation.

**Supplementary Materials:** The following supporting information can be downloaded at: https://www.mdpi.com/article/10.3390/rs15194693/s1. Text S1: Comparison between the site EC data and the experimental results; Table S1: Details of the three EC flux towers (CN-HaM, CF-HBG_S01, and CN-Ha2) in the Yellow River basin employed in this study. Note that GRA refers to grassland, and SHR refers to shrubland; Figure S1: Geographical location of the three EC flux towers (CN-HaM, CF-HBG_S01, and CN-Ha2) in the Yellow River basin employed in this study; Figure S2: Scatter plots between the observed EC GPP against MODIS monthly static and dynamic schemes. The blue, orange, and green points represent the CN-Ha2, CN-Ham, and CF-HBG_S01 sites, respectively. References [55–58] are cited in the supplementary materials.

**Author Contributions:** Conceptualization, J.J. and L.W.; methodology, software, validation, and formal analysis, W.H. and L.W.; resources, J.J. and S.W.; writing—original draft preparation, J.J., W.H. and L.W.; writing—review and editing, J.J., W.H., Y.W., Q.Z., X.F. and L.R.; visualization, W.H. and L.W. All authors have read and agreed to the published version of the manuscript.

**Funding:** This research was funded by the National Natural Science Foundation of China (U2243203, 41971374 and 41807173) and the Carbon Peak and Carbon Neutralization Key Science and Technology Program of Suzhou (ST202228).

**Data Availability Statement:** No applicable.

**Acknowledgments:** The authors would like to thank the following institutions or projects for generously providing the data used in this study: the United States Geological Survey (USGS), the European Space Agency (ESA), Google Earth Engine (GEE), FLUXNET, ChinaFLUX, and the North American Carbon Program (NACP) Multi-scale synthesis and Terrestrial Model Intercomparison Project (MsTMIP).

**Conflicts of Interest:** The authors declare no conflict of interest.

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
