# Peer review of "Non-Ignorable Differences in NIRv-Based Estimations of Gross Primary Productivity Considering Land Cover Change and Discrepancies in Multisource Products"

_remotesensing, doi:10.3390/rs15194693_

Round 1

Reviewer 1 Report

In this paper, titled “Potential outperformance of NIRv-derived gross primary productivity by considering land cover change and multisource products differentiation” (remotesensing-2582795), the authors discussed the influences on GPP estimation by NIRv, associated with both land cover change and differences in multisource products.

However, in the reviewer’s opinion, this paper (as current version) possibly had a little contribution to this filed. Mostly, major parts of this paper were conducted referring to the reference (“Tracking the seasonal and inter-annual variations of global gross primary production during last four decades using satellite near-infrared reflectance data” (Wang et al., 2021)). As the authors mentioned, being different from the reference (Wang et al., 2021), this paper investigated the influences on GPP estimation, mainly considered both land cover change and multisource products differentiation. By the way, the reviewer did not understand clearly what was “differetiation” mean? It might be not suitable, please check. In this paper, the authors just showed the differences in GPP estimated through NIRv among land cover products at different times (2018 vs. 2001) or from different sources (MCD12C1 from NASA vs. CCI from ESA). The GPP estimation largely depends on land cover type, as the models showed (Table 2). Accordingly, properties of the land cover product affect certainly the GPP estimation. Firstly, the accuracy of the land cover product is important. Then, as we know, there have been many land use and cover products, with different combination or comparisons many results on this kind of “difference” possibly might be observed (that is many other papers will be published without much value). In addition to just showing the influence on GPP estimation, the author should provide general users the message or advices for GPP estimation in practice.  

The authors should clearly show readers the significance of this paper, not just showing the comparison results. Furthermore, there are several obvious problems in this manuscript, which should be tackled or improved properly.

1, Mistakes in the main text (lines 116-120), “Research manuscripts reporting large datasets that are deposited in a publicly available database should specify where the data have been deposited and provide the relevant accession numbers. If the accession numbers have not yet been obtained at the time of submission, please state that they will be provided during review. They must be provided prior to publication.”.

2, As mentioned before, there have been many land use and cover products. Why the authors selected the MCD12C1 from NASA and CCI from ESA. As we know, the global products are always provided with low accuracy. Accordingly, results in land cover change from 2001-2018 might be not credible (3.3 Land cover change), which may affect GPP estimation and trend analysis followed by. Furthermore, both Figure 2 (for MCD12C1) and Figure 3 (for CCI) demonstrated the changes, however, there were no comparisons on them and the figure style was different. Why?

3, Why the AVHRR reflectance data were used?

4, The annotation for Figure 4 and Figure 5 possibly was wrong, “The panel (b-d) shows the geographical distribution of the difference between the GPP in the scheme â…¡/â…¢/â…£ andâ… , respectively”. Please check.

5, It seems similar (even same) annual trends are observed between (b) and (d) in Figure 6. Why?  

6, What did “outperformance” mean? In the reviewer’s opinion, the “outperformance” should be proved through estimation improvements as compared with ground truth. The comparison with other models or products (shown in Figure 9) is unconvincing.

Wish the comments above are valuable.

No comments.

Author Response

Response to Reviewer 1 Comments

In this paper, titled “Potential outperformance of NIRv-derived gross primary productivity by considering land cover change and multisource products differentiation” (remotesensing-2582795), the authors discussed the influences on GPP estimation by NIRv, associated with both land cover change and differences in multisource products.

[Answer] We greatly appreciate your general and specific comments, which are very important and helpful to improve our manuscript.

However, in the reviewer’s opinion, this paper (as current version) possibly had a little contribution to this filed. Mostly, major parts of this paper were conducted referring to the reference (“Tracking the seasonal and inter-annual variations of global gross primary production during last four decades using satellite near-infrared reflectance data” (Wang et al., 2021)). As the authors mentioned, being different from the reference (Wang et al., 2021), this paper investigated the influences on GPP estimation, mainly considered both land cover change and multisource products differentiation. By the way, the reviewer did not understand clearly what was “differetiation” mean? It might be not suitable, please check. In this paper, the authors just showed the differences in GPP estimated through NIRv among land cover products at different times (2018 vs. 2001) or from different sources (MCD12C1 from NASA vs. CCI from ESA). The GPP estimation largely depends on land cover type, as the models showed (Table 2). Accordingly, properties of the land cover product affect certainly the GPP estimation. Firstly, the accuracy of the land cover product is important. Then, as we know, there have been many land use and cover products, with different combination or comparisons many results on this kind of “difference” possibly might be observed (that is many other papers will be published without much value). In addition to just showing the influence on GPP estimation, the author should provide general users the message or advices for GPP estimation in practice. 

[Answer] Thanks for your questions and suggestions. Our research was conducted referring to Wang et al. (2021) and studied the influence of LC and LCC on estimating GPP by NIRv. The goal of this study is to evaluate the influences of LCC and DMP on NIRv-based GPP estimation and explore the corresponding uncertainties. Hence, we set up the static and dynamic schemes of LCC and the comparative experiments of different land cover products. We would to used "differentiation" to reflect the differences of different product classification systems. We might have misused this word and changed it to “difference” in the corresponding position. Our previous works have showed that there were great differences among different products (Wang et al., 2021a; Wang et al., 2021b). At present, the influence of LC has not been systematically considered in estimating GPP by NIRv. This was the starting point for us to consider land cover in the process of GPP estimation and the scientific issue that this paper paid attention to. Due to the lack of spatial observation data, we could not determine which land cover product is most in line with the actual land type distribution in the study area. However, we think it is necessary to consider LCC, which is better than static LC. The distribution and changing trend of LC are different among different LC products. We aim to point out that it was necessary to consider land cover changes and products when using NIRv to estimate GPP. At present, our present work just gives a clear signal that the LCC and the differences between different products need to be considered in the GPP estimation. In the future, we would conduct more detailed studies on the aspects to give practical suggestions for GPP estimation.

The authors should clearly show readers the significance of this paper, not just showing the comparison results. Furthermore, there are several obvious problems in this manuscript, which should be tackled or improved properly.

[Answer] Thanks for your suggestion. Our research wants to give a clear signal that it is necessary to consider land cover changes and products when using NIRv to estimate GPP. We added and highligthed that in the abstract and discussion.We tried our best to revise the entire manuscript according to your suggestions and comments, and some new points have been found. The revised parts have been marked in red in the manuscript, and the specific points addressed are given in the list below.

  1. Mistakes in the main text (lines 116-120), — “Research manuscripts reporting large datasets that are deposited in a publicly available database should specify where the data have been deposited and provide the relevant accession numbers. If the accession numbers have not yet been obtained at the time of submission, please state that they will be provided during review. They must be provided prior to publication.”.

[Answer] Sorry for the mistake. The corresponding part in the article has been corrected. Thanks.

  1. As mentioned before, there have been many land use and cover products. Why the authors selected the MCD12C1 from NASA and CCI from ESA. As we know, the global products are always provided with low accuracy. Accordingly, results in land cover change from 2001-2018 might be not credible (3.3 Land cover change), which may affect GPP estimation and trend analysis followed by. Furthermore, both Figure 2 (for MCD12C1) and Figure 3 (for CCI) demonstrated the changes, however, there were no comparisons on them and the figure style was different. Why?

[Answer] Thanks for your question. Due to the lack of spatial observation data, we could not determine which land cover product was most in line with the actual land type distribution in the study area. Our work only cross-contrasted the two land cover products (that is, MCD12C1 V6 and ESACCI), which were the widespread products and could be well matched with remote sensing GPP estimation. Moreover, the two products have been used in our previous researches and we had a comprehensive understanding of them. The goal of this study is to evaluate the influences of LCC and DMP on NIRv-based GPP estimation and explore the corresponding uncertainties. Figure 2 is the qualitative change of MODIS land cover and Figure 3 is the quantitative change of CCI land cover product. Both Figure 2 (for MCD12C1) and Figure 3 (for CCI) demonstrated the changes, however, there were no direct comparisons. Due to CCI only has the proportion of PTFs inside each pixel and does not define the specific type of pixel, there is no way to draw the spatial distribution map of land cover of CCI product. Therefore, Figure 3 took the LCC distribution of MCD12C1 as the base map, and the histogram and circular rings indicated the proportion of the land cover transition of CCI, which could show that the LCC of the two products was different. Figure 2 and Figure 3 have has been redrawn for better presentation.

  1. Why the AVHRR reflectance data were used?

[Answer] Thanks for your question. The major parts of this paper were conducted referring to the reference (“Tracking the seasonal and inter-annual variations of global gross primary production during last four decades using satellite near-infrared reflectance data” (Wang et al., 2021)). The NIRv-GPP relationships proposed by Wang et al. were used to estimate GPP for the different land cover types in this study. In order to make the formula of estimating GPP based on NIRv available, we used the same AVHRR reflectance data to calculate the monthly NIRv.

  1. The annotation for Figure 4 and Figure 5 possibly was wrong, —“The panel (b-d) shows the geographical distribution of the difference between the GPP in the scheme â…¡/â…¢/â…£andâ… , respectively”. Please check.

[Answer] Sorry for the mistake. The annotations for Figure 4 and Figure 5 have been revised. Thanks.

  1. It seems similar (even same) annual trends are observed between (b) and (d) in Figure 6. Why?  

[Answer] Thanks for your question. We rechecked the data and redrawn Figure 6. Sorry for the mistake. The annual trends of the new Figure 6 (b) had some changes and showed some differences with (d), mainly in the southeast of the study area.

  1. What did “outperformance” mean? In the reviewer’s opinion, the “outperformance” should be proved through estimation improvements as compared with ground truth. The comparison with other models or products (shown in Figure 9) is unconvincing.

[Answer] Thanks for your question and suggestion. We collated eddy-covariance (EC) observed daily gross primary productivity (GPP) as the field observations to investigate the outperformance of our results and added the comparison between the observed EC GPP against MODIS monthly static and dynamic schemes in the supplementary materials. The geographical location of the three EC flux towers was shown in Figure S1 and the validation scatter plots were shown in Figure S2. Thanks again for all of your comments and suggestions! Your valuable comments and suggestions have made great contributions to improving the content of our paper.

References:

  1. Wang, S.H.; Zhang, Y.G.; Ju, W.M.; Qiu, B.; Zhang, Z.Y. Tracking the seasonal and inter-annual variations of global gross primary production during last four decades using satellite near-infrared reflectance data. Sci. Total Environ. 2021a, 755, 142569. https://doi.org/10.1016/j.scitotenv.2020.142569
  2. Wang, L.H.; Jin, J.X. Uncertainty Analysis of Multisource Land Cover Products in China. Sustainability 2021b, 13, 8857. https://doi.org/10.3390/su13168857

Reviewer 2 Report

In general terms, the paper is organized well and is easy to follow.  However, I do present some comments and suggestions below for improvement. 

1.  Lines 116-120: These sentences have nothing to do with this research.  Are they part of this paper? 

2.  Figure 1: There are obvious errors in the frequency histogram, i.e., the sum of the ratio of grasslands and croplands has significantly exceeded 1. 

3.  Line 150: To be consistent with MCD12C1 resolution, the ESACCI data should be summarized into 0.05 degree instead of 0.5 degree. 

4.  In the section of "3.1 Land cover change", the author used MCD12C1 data for the analysis of the whole basin, while chose ESACCI data for the analysis of sub-basins.  Why not use a unified data source for separate analysis?  The current analysis results are confusing, MCD12C1 results show a decrease in grassland and an increase in cropland, while ESACCI results show an increase in grassland.  This diametrically opposed result requires explanation.  In addition, how do we know the changes of cropland in ESACCI analysis?

Author Response

Response to Reviewer 2 Comments

In general terms, the paper is organized well and is easy to follow. However, I do present some comments and suggestions below for improvement.

[Answer] We greatly appreciate your general and specific comments, which are very important and helpful to improve our manuscript. We tried our best to revise the entire manuscript according to your suggestions and comments, and some new points have been found. The revised parts have been marked in red in the manuscript, and the specific points addressed are given in the list below. 

  1. Lines 116-120: These sentences have nothing to do with this research.  Are they part of this paper? 

[Answer] Sorry for the mistake. The corresponding part in the article has been corrected. Thanks.

  1. Figure 1: There are obvious errors in the frequency histogram, i.e., the sum of the ratio of grasslands and croplands has significantly exceeded 1. 

[Answer] Thanks for your suggestion. In the frequency histogram of Figure 1, we set a discontinuity in the ordinate, which makes the sum seem unreasonable. We redrew Figure 1 to make its information more direct.

  1. Line 150: To be consistent with MCD12C1 resolution, the ESACCI data should be summarized into 0.05 degree instead of 0.5 degree. 

[Answer] Sorry for the mistake. The “0.5” has been revised to “0.05”. Thanks.

4.In the section of "3.1 Land cover change", the author used MCD12C1 data for the analysis of the whole basin, while chose ESACCI data for the analysis of sub-basins.  Why not use a unified data source for separate analysis?  The current analysis results are confusing, MCD12C1 results show a decrease in grassland and an increase in cropland, while ESACCI results show an increase in grassland.  This diametrically opposed result requires explanation.  In addition, how do we know the changes of cropland in ESACCI analysis?

[Answer] Thanks for your question. Figure 2 is the qualitative change of MODIS land cover and Figure 3 is the quantitative change of CCI land cover product. Both Figure 2 (for MCD12C1) and Figure 3 (for CCI) demonstrated the changes, however, there were no direct comparisons. We recalculated the changes of land cover types in CCI during 2001-2018 and found the grassland also decreased, which is consistent with the researches of Ji et al. (2021) and Tian et al., (2021). However, due to the difference of resolution and classification system, land cover types and changes of the two products were different. Figure 3 has been redrawn and added the changes of cropland. Thanks again for all of your comments and suggestions! Your valuable comments and suggestions have made great contributions to improving the content of our paper.

References:

  1. Wang, S.H.; Zhang, Y.G.; Ju, W.M.; Qiu, B.; Zhang, Z.Y. Tracking the seasonal and inter-annual variations of global gross primary production during last four decades using satellite near-infrared reflectance data. Sci. Total Environ. 2021, 755, 142569. https://doi.org/10.1016/j.scitotenv.2020.142569
  2. Ji, Q.L.; Liang, W.; Fu, B.J.; Zhang, W.B.; Yan, J.W.; Lü, Y.H.; Yue, C.; Jin, Z.; Lan, Z.Y.; Li, S.Y.; et al. Mapping land use/cover dynamics of the Yellow River Basin from 1986 to 2018 supported by Google Earth Engine. Remote Sens. 2021, 13, 1299. https://doi.org/10.3390/rs13071299
  3. Tian, F.; Liu L.Z.; Yang, J.H.; Wu, J.-H. Vegetation greening in more than 94% of the Yellow River Basin (YRB) region in China during the 21st century caused jointly by warming and anthropogenic activities. Ecol. Indic. 2021, 125, 107479. https://doi.org/10.1016/j.ecolind.2021.107479.

Round 2

Reviewer 1 Report

Thanks. The reviewer appreciated the authors' efforts and responses.  There still are several concerns. Why AVHRR reflectance was mainly used? When MODIS land use product was used, the MODIS reflectance might be more suitable. The authors discussed both land use/cover dynamics and different products on GPP estimation. However, there are still no solid  suggestions for data users, and just comparisons among different data combinations were shown. Accordingly, the "outperformance" may not be appropriate. As shown in Figure 9, median differences in GPP were larger between land use /cover products (MODIS vs ESA CCI) than between different years (i.e. dynamics). 

No comments.

Author Response

Comments 1: Thanks. The reviewer appreciated the authors' efforts and responses. There still are several concerns. Why AVHRR reflectance was mainly used? When MODIS land use product was used, the MODIS reflectance might be more suitable.

Response 1: We greatly appreciate your comments, which are very important and helpful to improve our manuscript. The methods of this work mainly referred to that of Wang et al. (2021, titled as “Tracking the seasonal and inter-annual variations of global gross primary production during last four decades using satellite near-infrared reflectance data”). In the work of Wang et al., the long-term observations of AVHRR reflectance were used to estimate the global GPP during 1982-2018. We used the same parameters of NIRv-GPP relationships of Wang et al. (2021) to estimate GPP for the different land cover types. In order to make the formulas work, we used the same AVHRR reflectance data to calculate the monthly NIRv.

Comments 2: The authors discussed both land use/cover dynamics and different products on GPP estimation. However, there are still no solid suggestions for data users, and just comparisons among different data combinations were shown.

Response 2: We aim to point out that it is necessary to consider land cover changes and the difference of multisource products when using NIRv to estimate GPP. Our present work has proved that considering the land cover change and the difference of multisource products would bring significant differences to GPP estimation by NIRv. While due to the lack of continuous observed spatial data, we just collated three eddy-covariance (EC) observed monthly GPP as the field observations to show the importance of considering land cover. At present, we give a clear signal that the LCC and the differences between multisource products need to be considered in the GPP estimation. In the future, we would conduct more detailed studies on the aspects to give practical suggestions for GPP estimation.

Nevertheless, three solid suggestions could be helpful for data users according to this study:

(i) The dynamic schemes had better performances than the static schemes. Considering the LCC when calculating GPP can deduce the uncertainties.

(ii) The LCC and the differences between different products need to be considered in the GPP estimation.

(iii) DMP exerts a more significant influence on GPP estimation, and both DMP and LCC's impacts on GPP estimation are non-ignorable for data users.

Comments 3: Accordingly, the "outperformance" may not be appropriate. As shown in Figure 9, median differences in GPP were larger between land use /cover products (MODIS vs ESA CCI) than between different years (i.e. dynamics).

Response 3: Considering your suggestion and the content of the article comprehensively, we revised the title of this paper to “Non-ignorable differences in NIRv-based estimations of gross primary productivity considering land cover change and discrepancy of multisource products”, and corresponding descriptions have been also revised throughout the article. Thanks again for all of your comments and suggestions!